# Thermal Time as a Parameter to Determine Optimal Defoliation Frequency of Perennial Ryegrass (*Lolium perenne* L.) and Pasture Brome (*Bromus valdivianus* Phil.)

Iván Calvache [1], Oscar Balocchi [1,*], Máximo Alonso [1], Juan Pablo Keim [1] and Ignacio F. López [2]

[1] Instituto de Producción Animal, Facultad de Ciencias Agrarias, Universidad Austral de Chile, PO Box 567, 5090000 Valdivia, Chile; ivan.calvache@uach.cl (I.C.); maximo.alonso@uach.cl (M.A.); juan.keim@uach.cl (J.P.K.)

[2] School of Agriculture and Environment, Massey University, Palmerston North 4442, New Zealand; I.F.Lopez@massey.ac.nz

\* Correspondence: obalocch@uach.cl; Tel.: +56-63-222-1242

**Abstract:** The herbage mass and nutritional value of harvested forage are fundamental determinants of the production potential of pastoral systems. The objective of this study was to evaluate the growth dynamics and accumulated herbage mass expressed in dry matter (DM) of perennial ryegrass (*Lolium perenne* L.) and pasture brome (*Bromus valdivianus* Phil.) pastures, using thermal time (TT) as a defoliation criterion. Thirty plots (15 of *L. perenne* and 15 of *B. valdivianus*) were distributed in three field blocks and subjected to five defoliation frequencies (DF) determined by TT, expressed as the accumulated growing degree-days (AGDD; DF1 = 90, DF2 = 180, DF3 = 270, DF4 = 360, and DF5 = 450 AGDD) for one year (2016), at the Austral Agricultural Experimental Station of the Universidad Austral de Chile. Every three days, the total leaf length (TLL) was measured, and the leaf elongation rate (LER, cm $L^{-1}$), leaf growth rate (LGR, cm $L^{-1}$), leaf appearance rate (LAR, d $L^{-1}$), phyllochron (AGDD $L-1$), and accumulated herbage mass per hectare (kg DM $ha^{-1}$) were calculated. Defoliations were scheduled according to AGDD, and a sample was taken from each cutting to determine (dry matter 'DM', crude protein 'CP', neutral detergent fiber 'NDF', acid detergent fiber 'ADF', water-soluble carbohydrates 'WSC' and metabolizable energy 'ME'). The pastures that were allocated to DF5 presented higher DM yields (12,600 kg DM $ha^{-1}$ $year^{-1}$), TLL (54.6 cm), and LER (0.63 cm $d^{-1}$) compared to pastures with high DF (90 and 180 ADGG). *B. valdivianus* presented a lower phyllochron than *L. perenne* (74.4 vs 87.9 AGDD $L^{-1}$, respectively). Concentrations of CP and ME decreased from the shortest DF (90 AGDD) to the largest DF (450 AGDD), dropping from 221 to 138 g $kg^{-1}$ CP and from 2.6 to 2.4 Mcal $kg^{-1}$ DM of ME. All variables were affected by the season (Ssn) ($p < 0.001$). The AGDD can be used as a defoliation criterion and a tool to balance yield with nutritive value according to the farmer's needs.

**Keywords:** growth dynamic; phyllochron; accumulated growing degree-days

## 1. Introduction

Defoliation frequency (DF) is an important factor that influences recovery of pastures to grazing events [1]. The purpose of controlling DF is to let pasture species (Sp) recover water-soluble carbohydrate (WSC) reserves in the roots and tillers, and therefore maximize the post-defoliation regrowth rate. Several studies reported that DF affects pasture dry matter (DM) yield, regrowth rate, and nutritive

value. Frequent defoliation has been shown to reduce persistence of perennial ryegrass, DM yield, and nutritive value when compared with infrequent defoliation [2,3].

The phyllochron has been defined as thermal time (TT) interval between the appearance of two successive leaves from the same stem [4]. The phyllochron value is measured in accumulated heat units, a value that remains constant under different daily temperature regimes [5]. In cool-season grasses, temperature is one of the main atmospheric factors that influences pasture growth and development, especially in Sp that need to accumulate chilling hours to progress from the vegetative to reproductive stage [6]. The plant physiological response to temperature varies according to Sp and cultivar [7]. For *Lolium perenne* L. (*L. perenne*), the phyllochron expressed as accumulated growing degree-days (AGDD) remains constant through time but varies seasonally and annually when expressed in "d leaf$^{-1}$". In spring and summer, the temperature accumulation can be as much as three times greater than in the winter months; hence, the phyllochron expressed in number of days varies significantly [8,9]. Luminous intensity, mineral nutrition, and water availability are atmospheric factors that also influence pasture growth and development, but to a lesser extent than the phyllochron [10]. Bartholomew [11], Davidson et al. [12] and Thiesen et al. [13] showed that the phyllochron is usually related to the phenological age of the plant and TT. As the growth cycle progresses, the accumulation of TT intensifies, and the number of leaves emerging from the same stem is closely correlated to AGDD [14]; thus, the phyllochron expressed as TT may be less variable than 'd leaf$^{-1}$' as a defoliation criterion, and therefore it would be a useful tool for grazing management decisions. AGDD is calculated according to the maximum and minimum daily temperatures and a base temperature [15]. Even though DF recommendations for perennial ryegrass have been established, in terms of the leaf stage [16,17], the literature has shown that the optimal DF can vary among pasture Sp [2,18]. Pasture brome, *Bromus valdivianus* Phil. (*B. valdivianus*), is a native grass from southern Chile that has a similar foliage mass and foliage quality to perennial ryegrass [19] and competes with perennial ryegrass in permanent pastures of humid temperate regions [20]. In Chile, growth dynamics pasture there is a lack of information on an optimal DF for this Sp. However, growth dynamics differ between the Sp; the *B. valdivianus* has 2.3 times heavier tillers, 3.6 times greater leaf area per tiller, 13% greater live leaf number per tiller, and 1.5 times longer total lamina length per tiller than *L. perenne* [21]. However, the accumulated herbage mass per year is similar between pasture brome and perennial ryegrass because the perennial ryegrass produces 2.2 times more tillers than pasture brome under Chilean conditions [19]. Both species are perennials, which allows them to survive if there are adequate environmental and management conditions. Therefore, the aims of this study were to evaluate the use of TT as a DF criterion and investigate the effects of DF on growth dynamics, accumulated herbage mass, and nutritive value of *L. perenne* L. and *B. valdivianus* Phil. throughout the year.

## 2. Materials and Methods

### 2.1. Site Description

The field site was located at the Agriculture Research Station (AARS) of the Universidad Austral de Chile (39°46' S, 73°13' W, 12 m above sea level) in Valdivia, southern Chile, from spring 2015 to autumn 2017. The climate is temperate with an average annual temperature of 12.5 °C and total annual rainfall of 1284 mm for 2016. This study included data from December 2015 to January 2017. Weather data were recorded at a meteorological station located 5 m from the field site. The mean monthly temperature and rainfall for the experiment period are shown in Figure 1. The soil is derived from volcanic ashes, classified as a Duric Hapludand (Valdivia Series according to CIREN, [22]), and the experimental site was located on a flat topography with slope less than 2%. Soil properties measured at the beginning of the experiment included pH 5.4, 1.46 g kg$^{-1}$ organic matter, 17.4 mg kg$^{-1}$ Olsen-P, and 4.9% aluminum saturation (Soil Lab, Institute of Agricultural Engineering and Soils, Faculty of

Agrarian Sciences, Universidad Austral de Chile, Valdivia, Chile, 2015); this allows it to be a suitable soil for perennial species.

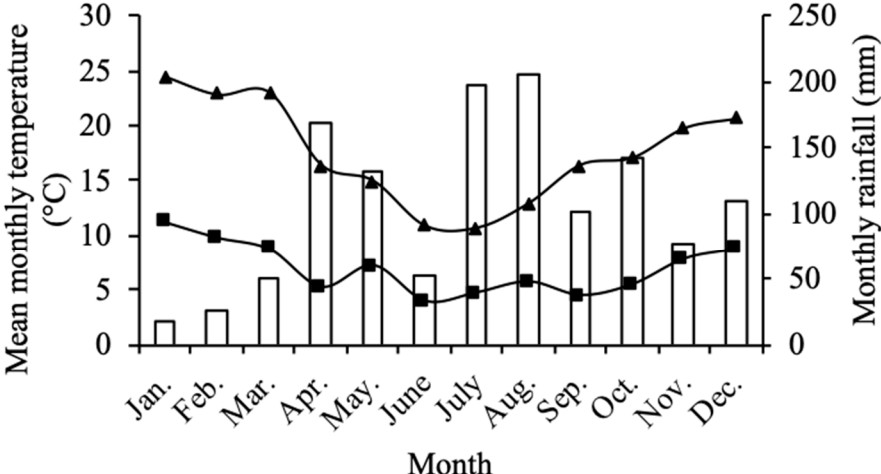

**Figure 1.** Mean monthly maximum (▲) and minimum (■) temperature and monthly rainfall (□) for Valdivia, Chile, from January to December 2016.

### 2.2. Experimental Design

Thirty 15 m$^2$ (3 × 5 m) plots were randomly distributed in three field blocks. Each block was composed of 10 plots: 5 sown with 30 kg ha$^{-1}$ *L. perenne* cv. Alto and 5 with 45 kg ha$^{-1}$ *B. valdivianus* Phil. native seed collected in the region. Plots were sown in autumn 2015 (April), with the first defoliation event during September 2015. The evaluation period started in December of the same year. Each plot was defoliated under one of the following fixed defoliation frequencies (DF) based on AGDD: DF1 = 90, DF2 = 180, DF3 = 270, DF4 = 360, and DF5 = 450 AGDD. Treatments corresponded to the interaction between the pasture Sp, DF, and season (Ssn).

### 2.3. Evaluated Variables and Sampling

In each plot, three tillers were marked at the base with a different color clip to identify them individually. The TLL was measured every three days and the appearance of new leaves was recorded. The TLL was measured for fully expanded leaves from the tip of the lamina to its ligule, while for growing leaves, lamina length was measured from the tip of the lamina to the ligule of the previously fully expanded leaf. The calculated variables during the growth dynamics measurement period on marked tillers were: Leaf appearance rate (LAR; leaves/day):

$$LAR = (n - 1)/t, \tag{1}$$

where $n$ = number of appearing leaves, and t = period of time between the appearance of the first and last leaf [8]. A new leaf was considered as "appeared" when its tip was visible [4]. The leaf elongation rate (LER; cm/day):

$$LER = [((l_2 - l_1)/t_1) + ((l_3 - l_2)/t_2) + \ldots + ((l_n - l_{n-1})/t_{n-1})]/(n-1), \tag{2}$$

where l = leaf blade length at the nth measurement, t = time between two consecutive measurements, and n = number of leaf elongation measurements performed. Finally, phyllochron in AGDD leaf$^{-1}$ (AGDD L$^{-1}$) was calculated as the accumulation of mean daily temperature; daily máximum ($T_{max}$) +

daily mínimum ($T_{\min}$)/2 above a base temperature ($T_{base}$) of 5 °C [23] for both Sp, so that accumulated temperature [24] over the period day 1 to day *n* is

$$\sum_{1}^{n}[(\mathrm{T}max + \mathrm{T}min)/2] \; / \; T\text{base} \qquad (3)$$

Air temperature at 150 cm above ground level was measured at a weather station adjacent to the field experiment.

When a pasture reached its corresponding DF, individual plots were harvested using a rotary mower equipped with a collection bag and a residual height of 5 cm was left. The herbage mass that was harvested and weighed corresponded to fresh tissue that grew during AGDD for each DF. From this fresh pasture, a 500 g sample was taken and oven dried at 60 °C for 72 h, or until reaching constant weight between consecutive weight measurements. The DM (g kg$^{-1}$) was calculated and applied to the rest of the harvested pasture from that plot. The dried herbage was ground in a Wiley mill using a 1 mm sieve (Model Digital ED-5, Thomas Scientifics, Swedesboro, NJ, USA) and then stored in plastic bottles with hermetic closures for subsequent nutritional value analysis.

To avoid any edge plot effect, at each defoliation event the 0.5 m edge strip of each plot was mowed and discharged, after which the pasture herbage mass contained in the rest of the plot (11.25 m$^2$) was harvested and evaluated. The seasonal and annual accumulated herbage mass were calculated for each plot, as the weight of all the cuts during the period.

### 2.4. Pasture Nutritive Value

The crude protein (CP), neutral detergent fiber (NDF), acid detergent fiber (ADF), metabolizable energy (ME), and water-soluble carbohydrate (WSC) were estimated by near-infrared spectroscopy technique (NIRS) with a FOSS-NIRSystems MODEL 6500 (FOSS NIRSystem Inc., Silver Spring, MD, USA) using the prediction equations developed based on the wet chemistry results developed by the Animal Nutrition Laboratory of the Universidad Austral de Chile. The standard errors of cross-validation were 0.78, 1.92, 1.19, 0.30, and 6.99 for CP, NDF, ADF, ME, and WSC, respectively, while the $R^2$ values were 0.98, 0.93, 0.93, 0.84, and 0.96, respectively. The growth dynamics data and nutrient concentration measurements were reported as the weighted average per DF, Sp, and Ssn and calculated as described by Deak et al. [25]:

$$\mathrm{WAX} = (\sum X_i \times HM_i)/\sum HM_i, \qquad (4)$$

where WAX represents the weighted average of X, $X_i$ is the nutrient concentration harvest i, and HMi corresponds to the herbage mass of harvest i.

### 2.5. Statistical Analysis

The study was established as a randomized block design with a factorial arrangement of the treatments considering the interaction between five DF (90, 180, 270, 360, and 450 AGDD), two grass Sp (*L. perenne* and *B. valdivianus*), and the four seasons of the year (summer, autumn, winter, and spring), repeated in three blocks.

The results were analyzed using the adjusted model PROC GLMMIX in SAS V9.0 (SAS Institute, Cary, NC, USA). When significant effects were detected, the PDIFF command was performed to separate treatment means with a confidence level of 95% using the Tukey option.

## 3. Results

### 3.1. Growth Dynamics

The accumulated herbage mass (kg DM ha$^{-1}$) and components of the yield as affected by DF, Sp, and Ssn are described in Table 1. There were significant effects for DF, Sp, and Ssn on TLL, LGR, and accumulated herbage mass ($p < 0.05$). The interactions between Sp × Ssn for the variables TLL, LGR, LAR, and accumulated herbage mass were significant and the interaction between DF × Ssn also was significant for LGR, TLL, and accumulated herbage mass.

**Table 1.** Growth dynamics and accumulated herbage mass of two species (*Lolium perenne* L. and *Bromus valdivianus* Phil.) as affected by five defoliation frequencies and season.

| DF (AGDD) | TLL (cm) | LER (cm d$^{-1}$) | LAR (d L$^{-1}$) | Phyllochron (AGDD L$^{-1}$) | Accumulated Herbage Mass (kg DM ha$^{-1}$ Year$^{-1}$) |
|---|---|---|---|---|---|
| 1 (90) | 36.09 [d] | 0.49 [c] | 15.05 | 79.21 | 7576 [e] |
| 2 (180) | 38.24 [cd] | 0.51 [bc] | 15.32 | 79.23 | 8612 [d] |
| 3 (270) | 46.11 [bc] | 0.53 [bc] | 15.97 | 78.67 | 10463 [c] |
| 4 (360) | 51.24 [ab] | 0.59 [ab] | 15.32 | 82.79 | 11432 [b] |
| 5 (450) | 54.61 [a] | 0.63 [a] | 15.87 | 85.96 | 12600 [a] |
| m.s.e. | 2.055 | 0.020 | 0.611 | 2.320 | 227.264 |
| *p*-Value | <0.0001 | <0.0001 | 0.1132 | 0.1319 | <0.0001 |
| **Sp** | | | | | |
| *L. perenne* | 38.44 | 0.47 | 16.35 | 87.9 | 9497 |
| *B. valdivianus* | 52.47 | 0.63 | 14.66 | 74.44 | 10760 |
| m.s.e. | 1.299 | 0.013 | 0.386 | 1.467 | 35.934 |
| *p*-Value | <0.0001 | <0.0001 | <0.0001 | <0.0001 | <0.0001 |
| **Ssn** | | | | | **Accumulated Herbage Mass (kg DM ha$^{-1}$ Ssn$^{-1}$)** |
| Summer | 10.51 [d] | 0.19 [d] | 11.20 [a] | 86.00 [ab] | 1863 [c] |
| Autumn | 60.12 [b] | 0.68 [b] | 16.80 [c] | 81.90 [b] | 2371 [b] |
| Winter | 42.12 [c] | 0.45 [c] | 21.49 [d] | 72.86 [c] | 1440 [d] |
| Spring | 69.08 [a] | 0.87 [a] | 12.53 [b] | 92.87 [a] | 4454 [a] |
| m.s.e. | 1.838 | 0.018 | 0.547 | 2.075 | 50.818 |
| *p*- Value | <0.0001 | <0.0001 | <0.0001 | <0.0001 | <0.0001 |
| **Interaction** | | | ***p*-Value** | | |
| Ssn × Sp | 0.0063 | <0.0001 | <0.0001 | 0.0821 | <0.0001 |
| Ssn × DF | 0.1256 | <0.0001 | 0.2451 | 0.0504 | <0.0001 |
| Sp × DF | 0.7237 | 0.9235 | 0.9397 | 0.6682 | 0.3845 |
| DF × Sp × Ssn | 0.5962 | 0.7706 | 0.2191 | 0.6141 | 0.4024 |

DF: Defoliation frequency measured as accumulated growing degree-days (AGGD), Ssn: season, Sp: species, TLL: Total leaf length, LER: Leaf elongation rate, LAR: Leaf appearance rate, m.s.e.: standard error of the mean. Different letters in the same column indicate significant differences according to LSD at the 95% confidence level.

The largest TLL (54.6 cm) was measured in DF5 (450 AGDD) and was statistically longer than DF3 (270 AGDD) and DF1 (90 AGDD) by 18.43% and 51.3%, respectively. The greatest LAR (14.6 d L$^{-1}$) was also measured in DF5, while phyllochron was not affected by DF. Annual accumulated herbage mass was 13% greater for *B. valdivianus* (10,760 kg DM ha$^{-1}$) compared with *L. perenne* (9,497 kg DM ha$^{-1}$), as well as TLL (+14 cm) and LER (+), whereas LAR and phyllochron were higher for *L. perenne* than *B. valdivianus*. In terms of growth patterns across seasons, the weather conditions changed from one Ssn to the next, affecting all of the evaluated variables: the TLL varied from 69.1 cm in spring to 10.5 cm in summer, and LER had a similar trend decreasing from 0.8 cm d$^{-1}$ in spring to 0.2 cm d$^{-1}$ during summer. The phyllochron varied according to Ssn, being lowest during winter (72.8 AGDD L$^{-1}$), highest in spring (92.8 AGDD L$^{-1}$) and summer (86.0 AGDD L$^{-1}$), followed by autumn (81.9 AGDD L$^{-1}$), although summer and autumn were not significantly different. The highest

herbage mass per Ssn occurred in spring with 4454 kg DM ha$^{-1}$ and the fastest LAR in summer with 11.2 d L$^{-1}$.

The Sp × Ssn interaction was statistically significant ($p < 0.05$) for TLL and herbage mass, with *B. valdivianus* being higher than *L. perenne* (Figure 2a,d) for all seasons. The LER presented differences only in autumn and spring, with *B. valdivianus* showing higher rates than *L. perenne* (0.8 vs. 0.55 cm d$^{-1}$ during autumn and 1.0 vs. 0.73 cm d$^{-1}$ during spring) (Figure 2b). In summer, LAR was the only evaluated variable that was statistically different between Sp, in which *B. valdivianus* required 9 days for new leaves to appear, while 14 days were required for *L. perenne* (Figure 2c).

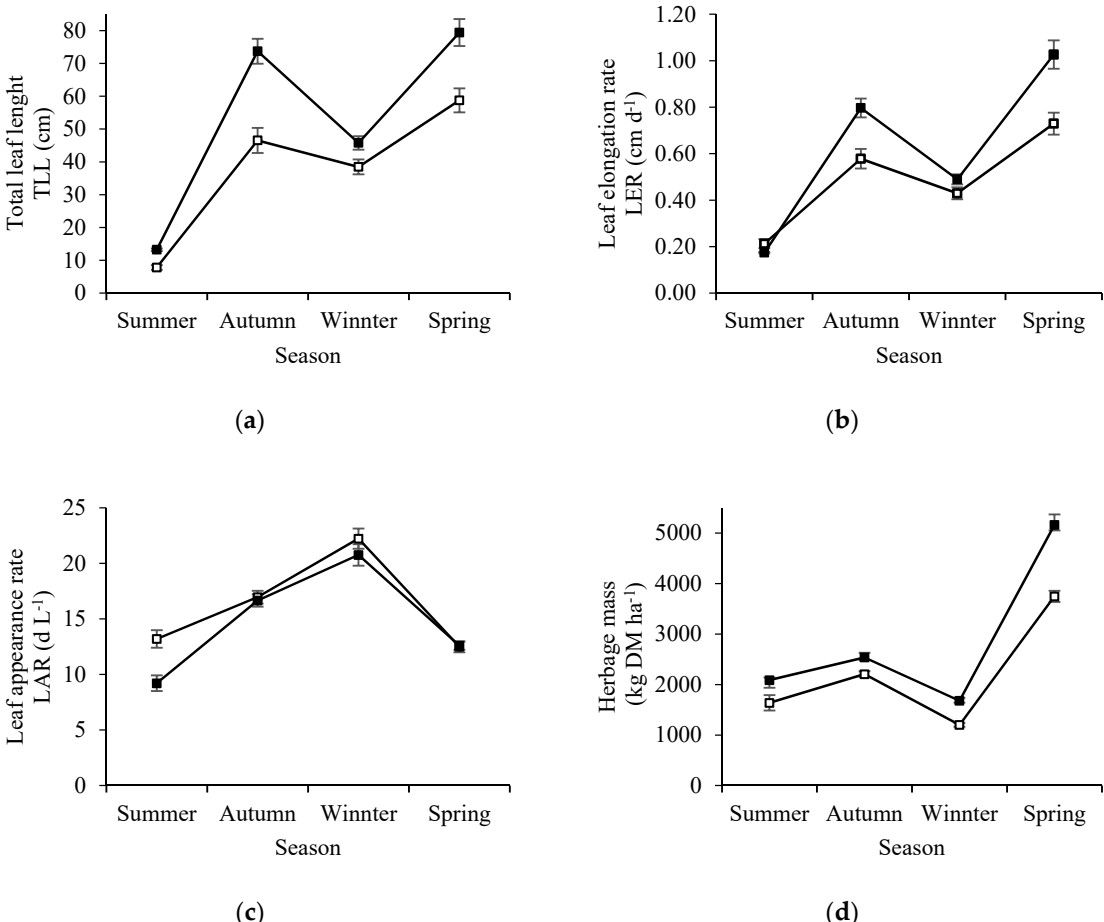

(**a**)　　　　　　　　　　　　　　　　　　　　　　　　(**b**)

(**c**)　　　　　　　　　　　　　　　　　　　　　　　　(**d**)

**Figure 2.** Interaction between season and species for: (**a**) total leaf length, (**b**) leaf elongation rate, (**c**) leaf appearance rate and (**d**) herbage mass in *Lolium perenne* L. (□) and *Bromus valdivianus* Phil. (■).

### 3.2. Nutritive Value

Table 2 reports the DF, Ssn, and Sp effects on nutrient concentrations. ME, CP, and NDF were affected by the interaction between DF × Sp × Ssn ($p < 0.001$).

In plots with a high DF (90 AGDD), the concentration of CP and DM was 37.8% and 4% higher, respectively, than in plots with a low DF (450 AGDD), and the concentration of NDF, ADF, and WSC was reduced by 15.2%, 24%, and 27%, respectively. In the Sp comparison, ME and WSC content was higher in *L. perenne* with 2.60 Mcal kg DM$^{-1}$ and 130 g kg$^{-1}$, respectively, and *B. valdivianus* had higher CP (189.3 g kg$^{-1}$), NDF (558.0 g kg$^{-1}$), and ADF (313.6 g kg$^{-1}$) (Table 2,). The change in ADF concentration through the seasons was consistent. The highest concentrations were measured with lower defoliation frequencies (360 and 450 AGDD). In contrast, the DM concentration was more influenced by the Ssn than by the DF, since summer presented the highest DM values across all DF.

**Table 2.** Nutritive value of two species (*Lolium perenne* L. and *Bromus valdivianus* Phil.) at five defoliation frequencies across seasons.

| DF (AGDD) | CP (g kg⁻¹) | ME (Mcal kg⁻¹ DM) | NDF (g kg⁻¹) | ADF (g kg⁻¹) | WSC (g kg⁻¹ DM) | DM (g kg⁻¹) |
|---|---|---|---|---|---|---|
| 1 (90) | 221.4 [a] | 2.58 [a] | 476.7 [c] | 250.9 [c] | 94.46 [c] | 249.4 [b] |
| 2 (180) | 203.8 [a] | 2.59 [a] | 496.9 [bc] | 267.2 [bc] | 103.22 [c] | 235.1 [b] |
| 3 (270) | 189.0 [ab] | 2.57 [a] | 516.9 [b] | 287.4 [b] | 112.03 [b] | 243.9 [b] |
| 4 (360) | 167.0 [b] | 2.54 [a] | 543.9 [a] | 317.1 [a] | 124.83 [a] | 250.6 [b] |
| 5 (450) | 137.7 [c] | 2.48 [b] | 562.5 [a] | 333.6 [a] | 129.99 [a] | 288.0 [a] |
| m.s.e. | 1.46 | 0.004 | 1.65 | 1.51 | 1.178 | 5.05 |
| *p*-Value | <0.0001 | <0.0001 | <0.0001 | <0.0001 | <0.0001 | <0.0001 |
| **Sp** | | | | | | |
| *L. perenne* | 178.3 | 2.60 | 480.0 | 268.9 | 123.90 | 246.7 |
| *B. valdivianus* | 189.3 | 2.50 | 558.8 | 313.6 | 98.91 | 260.1 |
| m.s.e. | 0.92 | 0.003 | 1.04 | 1.07 | 0.745 | 3.19 |
| *p*-Value | <0.0001 | <0.0001 | <0.0001 | <0.0001 | <0.0001 | 0.0037 |
| **Ssn** | | | | | | |
| Summer | 186.2 [b] | 2.39 [c] | 503.6 [b] | 293.7 [a] | 96.55 [c] | 410.4 [a] |
| Autumn | 236.8 [a] | 2.64 [a] | 518.4 [b] | 282.5 [ab] | 107.59 [b] | 174.0 [c] |
| Winter | 172.3 [b] | 2.60 [a] | 507.8 [b] | 279.0 [b] | 121.04 [a] | 159.9 [c] |
| Spring | 139.9 [c] | 2.56 [b] | 547.9 [a] | 309.8 [a] | 126.44 [a] | 269.4 [b] |
| m.s.e. | 1.30 | 0.004 | 1.48 | 1.69 | 1.054 | 4.51 |
| *p*-Value | <0.0001 | <0.0001 | <0.0001 | <0.0001 | <0.0001 | <0.0001 |
| **Interaction** | **p-Value** | | | | | |
| Ssn × Sp | <0.0001 | <0.0001 | <0.0001 | <0.0001 | <0.0001 | 0.0023 |
| Ssn × DF | <0.0001 | <0.0001 | <0.0001 | <0.0001 | <0.0001 | <0.0001 |
| Sp × DF | 0.0002 | 0.0002 | <0.0001 | 0.0022 | <0.0001 | 0.3453 |
| Ssn × Sp × DF | 0.0806 | <0.0001 | 0.0002 | 0.0792 | <0.0001 | 0.1996 |

DF: Defoliation frequency measured as accumulated growing degree-days (AGGD), Sp: species, Ssn: season, CP: crude protein, ME: metabolic energy, NDF and ADF: neutral and acid detergent fiber, respectively, WSC: water-soluble carbohydrates, DM: dry matter, m.s.e.: standard error of the mean. The different letters in the same column indicate significant differences inside the factor, $p < 0.05$: expresses significant differences according to LSD and 95% confidence interval.

Table 3, shows the interaction between Ssn and DF for CP, ADF, and DM. The highest concentration of protein was recorded in autumn, varying from 271.0 g kg⁻¹ at 90 AGDD to 221.4 g kg⁻¹ at 450 AGDD. There were no statistical differences ($p > 0.05$) between summer and winter and DF. The lowest CP concentrations were 80.5 g kg⁻¹ (450 AGDD) and 109.9 g kg⁻¹ (369 AGDD), measured in spring. The highest CP concentrations were observed in autumn with values declining consistently as DF were reduced (DF90 > DF180 > DF270 > DF360 > DF450), whereas in summer the CP concentration was similar for DF90, DF180, and DF270 but greater than DF360 and DF450; in winter no differences among DF270 and DF360 were observed, while in spring DF360 and DF450 were similar.

The interaction between the main effects (DF, Sp, and Ssn) was significant for ME, NDF, and WSC ($p < 0.05$, Table 3,). The highest ME concentration (2.72 Mcal kg DM⁻¹) corresponded to *L. perenne* plots during autumn with a DF of 90 AGDD, which was significantly different from DF of 450 AGDD (2.65 Mcal kg DM⁻¹) (Figure 3a, $p < 0.05$). The lowest ME concentrations were measured in summer for both *B. valdivianus* and *L. perenne*. Differences in ME between Sp were significant ($p < 0.05$) for DF from 90 to 360 AGDD for all seasons, but interspecies differences were not significant for 450 AGDD. In both forage Sp it was observed that the concentration of NDF increased as the growth periods were extended, varying from 42% for *L. perenne* in spring at 90 AGDD to 64% for *B. valdivianus* at 450 AGDD in the same Ssn. The latter was observed for ADF concentration. The concentration of WSC for *L. perenne* ranged from 117.4 g kg⁻¹ to 180 g kg⁻¹ and was higher in all seasons at all DF than

*B. valdivianus* (Figure 3c). The lowest concentration was measured under high DF (90 AGDD) and increased with decreasing DF.

**Table 3.** Interaction between season and defoliation frequency by nutritional value.

| Nutrient | Ssn/DF | 90 | 180 | 270 | 360 | 450 | m.s.e. | *p*-Value |
|---|---|---|---|---|---|---|---|---|
| CP (g kg⁻¹) | Autumn | 271.6 a | 255.5 b | 238.9 c | 221.4 d | 196.1 fgh | 2.915 | <0.001 |
| | Winter | 216.0 de | 187.5 gh | 169.0 ij | 153.7 jk | 135.3 l | | |
| | Spring | 193.6 fgh | 166.2 j | 149.0 kl | 109.9 m | 80.5 m | | |
| | Summer | 204.2 ef | 206.1 def | 199.0 fg | 182.9 hi | 138.8 kl | | |
| ADF (g kg⁻¹) | Autumn | 243.4 l | 262.0 ijk | 283.9 fgh | 304.3 de | 318.5 cd | 3.381 | <0.001 |
| | Winter | 245.9 kl | 266.1 hij | 277.7 ghi | 299.9 ef | 305.2 de | | |
| | Spring | 250.2 jkl | 268.8 hi | 296.0 ef | 355.6 b | 378.2 a | | |
| | Summer | 263.9 ijk | 271.7 hi | 291.8 efg | 308.5 de | 332.3 c | | |
| DM (g kg⁻¹) | Autumn | 191.6 e | 153.9 e | 160.8 e | 181.7 e | 181.8 e | 10.092 | <0.001 |
| | Winter | 156.7 e | 161.2 e | 157.1 e | 164.4 e | 159.9 e | | |
| | Spring | 245.2 d | 245.8 d | 258.3 d | 258.6 d | 338.9 c | | |
| | Summer | 404.0 b | 379.3 bc | 399.3 b | 397.6 b | 471.4 a | | |

Ssn: season, DF: defoliation frequency (days), CP: crude protein, ADF: acid detergent fiber, DM: dry matter, m.s.e.: standard error of the mean. The different letters represent significant differences in the nutrient. *p* < 0.05: expresses significant differences according to LSD and 95% of confidence.

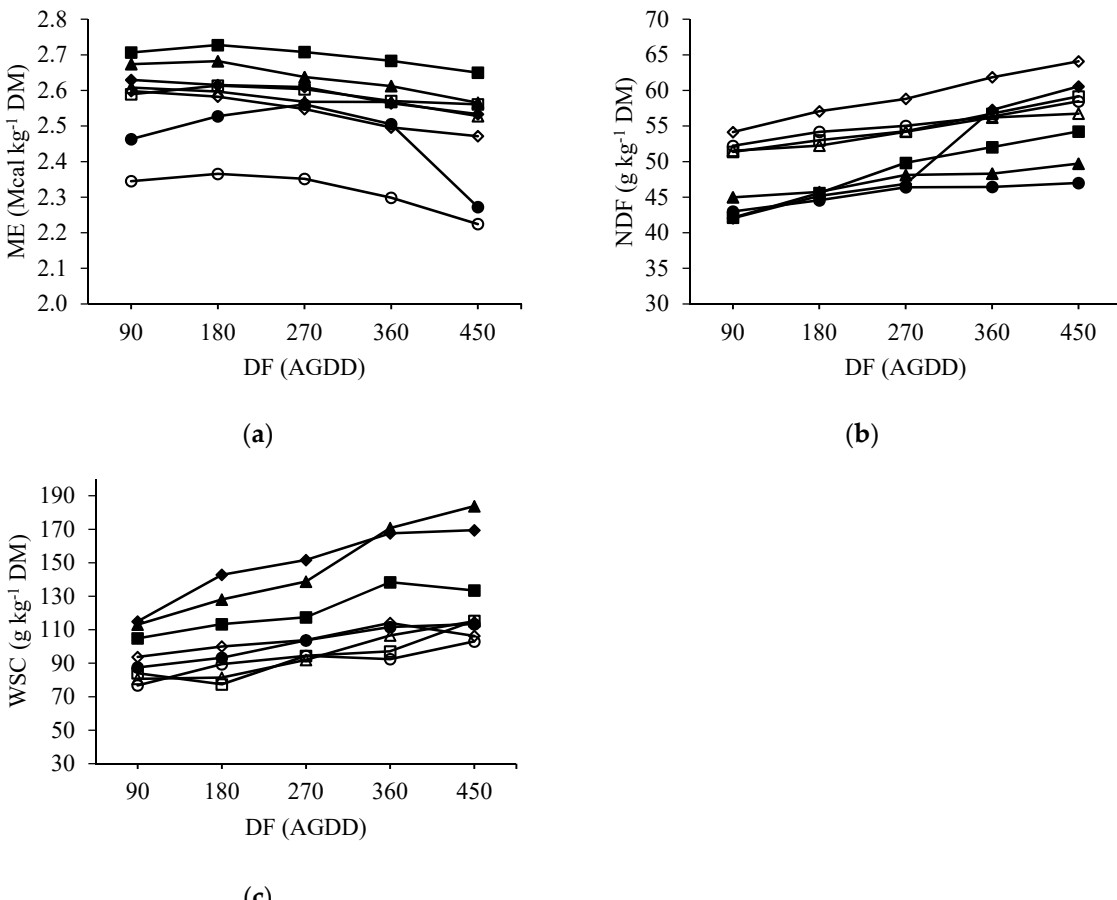

**Figure 3.** Interaction between defoliation frequency (DF) measured as accumulated growing degree-days (AGDD), species and season for: (**a**) metabolizable energy (ME), (**b**) neutral detergent fiber (NDF), and (**c**) water-soluble carbohydrates (WSC). *Lolium perenne* in summer (●), *Bromus valdivianus* in summer (○), *L. perenne* in autumn (■), *B. valdivianus* in autumn (□), *L. perenne* in winter (▲), *B. valdivianus* in winter (△), *L. perenne* in spring (♦) and *B. valdivianus* in spring (◊).

## 4. Discussion

### 4.1. Growth Dynamics

The results of this study show that growth, development, and nutritional value of both *L. perenne* and *B. valdivianus* can be modified by DF. The changes in growth attributes and nutritional value show how a shorter DF, and therefore a longer growth period and AGDD, influenced modifications in plants [26,27].

The differences in accumulated herbage mass, TLL, and LGR between DF are mostly associated with ability of plants to accumulate reserve carbohydrates that support its next regrowth [28,29]. This is supported by the WSC values for the different DF (Table 3), where, as DF increased, WSC concentration decreased. Studies have shown that in perennial ryegrass pastures defoliated after the third or fourth leaf emergence, the accumulated WSC are enough to sustain the next regrowth cycle [16,30,31].

On the other hand, when pastures are defoliated in the early stages of development (between 1 to 2 leaves), the amount of WSC is not sufficiently recovered to support the following regrowth, and the pasture yield diminishes [32]. High DF implies that an individual plant has to adapt its growth habits, through phenotypic plasticity, for its survival such that the stubble is shortened and by increasing the speed of WSC accumulation [33]. *Bromus valdivianus* outperformed *L. perenne* in variables such as accumulated herbage mass, TLL, and LER, which are closely related to the Sp' growth strategy. In the same time period, *B. valdivianus* had more leaves per tiller than *L. perenne* (Figure 2a), which explains its higher leaf extension rate and leaf appearance rate [34].

The lower TLL, LGR, LAR, and accumulated herbage mass during summer were probably associated with the low rainfall during that period, which was not able to fulfill the pasture soil water demands and was also affected by temperatures above the growth threshold. Guy et al. [35] reported that increase in temperatures above the growth threshold reduces the metabolic activity of pasture Sp, with special emphasis on carbohydrate partitioning, due to their importance as an energy source for cell expression. This has also been reported in other forage Sp (i.e., *Andropogon gerardii* Vitman, *Sorghastrum nutans* (L.) Nash, and *Schizachyrium scoparium* (Michx) Nash), as well as in some sorghum genotypes [11,36].

In this study, the phyllochron, expressed as AGDD, differed between seasons but not DF. This indicates that there is a constant phyllochron response, most likely determined by the genetics of the plant [37], but it can also be affected by chronological time depending on the Ssn and weather conditions and, to a lesser extent, by other aspects such as water availability and soil fertility [10]. This assumption is supported by Acharán et al. [3] and Balocchi et al. [1], which showed that the *L. perenne* phyllochron is not modified by DF or nitrogen fertilization but varies from autumn (May) to spring (November). This study shows that *B. valdivianus* required 13.5 AGDD for new leaf appearance, which affirms the theory that the phyllochron is very dependent on the phenological stage of each Sp and must be determined for each Sp [12,38].

The differences measured between the two Sp for variables such as accumulated herbage mass, TLL, and LAR under the same environmental conditions were expected due to their distinct growth habits. Similar results have been reported regarding the differences in growth dynamics of *L. perenne* and *B. stamineus* pastures and oat crops (*Avena sativa* L.), in which the temperature requirements for new leaf appearance were Sp specific, resulting in distinct growth patterns [39,40].

### 4.2. Nutritive Values

For nutritional values, most of the interactions between the main effects were significant, indicating that the effect of DF is dependent on the Sp and Ssn. The reduction in nutritive values represented by high concentrations of NDF and low ME and CP content in prolonged DF (450 AGDD), may be due to the accumulation of reproductive plant components and higher content of dead material [17]. All of this contributes to changes in the cells structure, increasing primary and secondary walls, and reducing the cellular content as the individual pasture plants and tillers enter into the reproductive stage [12,41].

The effect of DF on nutritional values in pastures and crops [18,27] are consistent with the present study, noting that, regardless of the defoliation criterion used (regrowth days, leaf condition, or number of leaves, among others), defoliated pastures at a state of advanced maturity (between three and five leaves) have lower DM and CP content with higher NDF concentrations than pastures defoliated at a higher frequency.

Although the nutritive value of *L. perenne* and *B. valdivianus* followed a similar pattern in relation to DF, CP and NDF concentrations were higher in *B. valdivianus* than *L. perenne*, but the latter presented higher ME and WSC values. These differences are best explained by the differences in LER, which is based on the number of leaves that each tiller can develop within a certain time, between the two Sp when defoliated at the same thermal time [42]. These results may also be attributable, in part, to the fact that growth and elongation rates of *B. valdivianus* were faster than *L perenne*. Berone et al. [43] showed that in the humid Pampas of Argentina *B. stamineus* (Roadside brome) has a higher leaf elongation rate per tiller than *L. perenne* at the same temperature. Similar results were reported by Turner et al. [18] when they evaluated the effect of DF from 2 to 4 leaves on *L. perenne*, *B. willdenowii* Kunth (Rescuegrass) and *Dactylis glomerata* L. (Orchardgrass) under greenhouse conditions. They reported that CP concentration varies significantly between Sp, but not between DFs within the same Sp. The same trend was reported for DM, with statistical differences for the interaction between Sp and DF.

The WSC concentration increased with decreasing DF (90 vs 450 AGDD), a result supported by Fulkerson and Donaghy [16] and Turner et al. [31] finding that the concentration of WSC in *L. perenne* was higher at the 4-leaf stage compared to the two-leaf stage. Based on these results, WSC concentration could be used as a plant recovery indicator for post-defoliation pasture recovery, since stores are depleted immediately upon defoliation and are only resynthesized once there are enough expanded leaves to support the growing energy demand [44].

The significant interaction between Ssn and DF for the nutritional variables indicates that the concentrations of CP and DM were greater in autumn and spring under high DF. Similar results were reported by [45] when they evaluated the effect of the time of year and nitrogen fertilization in *B. inermis* L. pastures, showing that protein content is maximized during the transition from winter to spring when plants are in a vegetative state.

## 5. Conclusions

The phyllochron was not affected by the DF, but it was influenced by the forage Sp, showing that *L. perenne* has a wider phyllochron (90 AGDD) than *B. valdivianus* (80 AGDD). The Ssn of the year influenced the phyllochron by making it shorter in times of low temperature.

The DF of 450 AGDD produced greater herbage mass accumulation than the shorter DF of 90 AGDD because tillers were able to grow longer leaves. The herbage mass accumulation increase due to a longer DF was accompanied by a decrease in nutritional value in *L. perenne* and *B. valdivianus*. The nutritional value of *L. perenne* and *B. valdivianus* defoliated at 90 and 180 AGDD was better with high concentration of ME and CP and a lower concentration of NDF and ADF, than with defoliation at 360 and 450 AGDD. The best balance between accumulated herbage mass and nutritive value for both Sp occurred with defoliation at 270 AGDD.

Consequently, under our experimental conditions AGDD showed to be a feasible alternative to be used as a defoliation criterion for *L. perenne* and *B. valdivianus* pastures. We encourage further research in other environments and with others grass species.

**Author Contributions:** Conceptualization: I.C., O.B., M.A., J.P.K. and I.F.L.; Data curation: I.C. and M.A.; Formal analysis: I.C., J.P.K. and I.F.L.; Funding acquisition: O.B.; Investigation, I.C., O.B. and J.P.K.; Methodology: I.C., O.B. and I.F.L.; Resources: O.B.; Supervision: M.A., J.P.K. and I.F.L.; Writing—original draft, I.C.; Writing—review & editing, O.B. All authors have read and agreed to the published version of the manuscript.

**Funding:** This research was funded by The National Fund for Scientific and Technological Development, Chile. Project Fondecyt 1180767.

**Acknowledgments:** Special thanks to Felipe Bozo, Carolina Valdes and Katheren Drummond who participated in the data collection. The first author obtained a CONICYT-PCHA Doctorado Nacional 2015 Scholarship towards his PhD program.

**Conflicts of Interest:** The authors declare no conflict of interest.

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
