# Peer review of "Thermal Time as a Parameter to Determine Optimal Defoliation Frequency of Perennial Ryegrass (Lolium perenne L.) and Pasture Brome (Bromus valdivianus Phil.)"

_agronomy, doi:10.3390/agronomy10050620_

Round 1
Reviewer 1 Report
The manuscript is well written and was very interesting to read. I have several comments for consideration.
- I had a difficult time in the M&M section as well as the results and discussion sections understanding if the authors were referring to initial growth, regrowth, or stockpiled forage. If this is clarified the other comments are minor. I think they were harvesting fresh stockpiled tissue at each DF. Yet they discussed adding DM together to get seasonal and annual forage.
- The authors need to go through the manuscript and make sure that the first time a Latin name is used it is associated with the authority, but not thereafter.
- Also, they need to do a better job at describing an abbreviation the first time it is used and then to be consistent in following the abbreviation and not spelling out the phrase.
Author Response
Response to Reviewer 1 Comments
Point 1: I had a difficult time in the M&M section as well as the results and discussion sections understanding if the authors were referring to initial growth, regrowth, or stockpiled forage. If this is clarified the other comments are minor. I think they were harvesting fresh stockpiled tissue at each DF. Yet they discussed adding DM together to get seasonal and annual forage.
Response 1: In M&M the clarification was made by saying that the forage measured and harvested corresponded to the vegetative material that grew during the AGDD for each of the defoliation frequencies. The results and discussion also clarified that forage accumulation was measured between one defoliation and another for the same defoliation frequency (Pp. 1, L 20 and 29., Pp. 2 L 55 and 75., Pp. 4, L 129, 130 and 164. and Pp. 5, L 188).
Point 2: The authors need to go through the manuscript and make sure that the first time a Latin name is used it is associated with the authority, but not thereafter.
Response 2: There was made a review of the entire manuscript and the advice for the case of Lolium perenne L. and Bromus valdivianus Phil. was adopted (Pp. 2, L 55 and 69).
Point 3: Also, they need to do a better job at describing an abbreviation the first time it is used and then to be consistent in following the abbreviation and not spelling out the phrase.
Response 3: The entire manuscript was reviewed and the abbreviations and their complete names were better described like in the case of the defoliation frequency (DF), Species (Sp) and season (Ssn) and were considered in the manuscript, but the complete name was left only in tables and charts.
Thank you to the reviewer for the valuable contribution.

Reviewer 2 Report
Paper is well written, very clear and well organized, indicating a consolidated expertise on the topic.
Just some minor suggestions and comments for authors
Introduction
Abbreviations in the introduction section should all be written out with acronyms in brackets as well as scientific names at the first mention.
Table 1. zero decimals in the last column (Herbage mass) or use t ha-1 as unit, instead of kg ha-1.
Material and method
2.1. Site Description
For northern hemisphere readers, please, add specific information on climate type and cumulative annual rainfall value to complement information from figure 1.
L 193 are you sure that ME and NDF were affected by a strong interaction between DF x Sp x Ssn (P < 0.001)? Please, verify.
- Please, give some additional details regarding the longevity of both perennial grass species in your environment/conditions.
- Based on 1), are the conclusions you presented always valid or it needs further studies over time (i.e. years subsequent to the sowing year). A comment on it should be appropriate and useful, for example at the end of discussion section or considered in conclusions.
In other words, is the (final) conclusion at line 334-335 valid/true or not for the years subsequent to the sowing year?
L 334 Consequently, AGDD is a feasible alternative to be used as a defoliation criterion for L. perenne and B. valdivianus pastures.
Author Response
Response to Reviewer 2 Comments
Point 1: Abbreviations in the introduction section should all be written out with acronyms in brackets as well as scientific names at the first mention.
Response 1: Changes to abbreviations were made in the introduction and throughout the manuscript (M&M, results and discussion), more precisely for the scientific names of forage species and some factors like defoliation frequency (DF), specie (Sp), season (Ssn), but the complete name was left in tables and charts.
Point 2: Table 1. zero decimals in the last column (Herbage mass) or use t ha-1 as unit, instead of kg ha-1.
Response 2: Significant numbers were used to approximate the decimals in the tables for accumulated herbage mass and, the values expressed in kg ha-1 were left (Pp. 5, table 1).
Point 3: Site Description. For northern hemisphere readers, please, add specific information on climate type and cumulative annual rainfall value to complement information from figure 1.
Response 3: In M&M, there was added that it is a temperate climate with an average temperature of 12.5 ° C and precipitation for 2016, which was the measurement period of the study with 1284 mm. (Pp. 2, L 87 and 88).
Point 4: L 193 are you sure that ME and NDF were affected by a strong interaction between DF x Sp x Ssn (P < 0.001)? Please, verify.
Response 4: The review was made, and it was decided to remove the word strong due to the P - value for the FDN was <.0002 in the DF x Sp x Ssn interaction (Pp. 6, L 201 and table 2).
Point 5: Please, give some additional details regarding the longevity of both perennial grass species in your environment/conditions.
Response 5: At the end of the introduction, it was clarified that these species have the characteristic perennials, which allows them to survive if there are adequate environmental and management conditions (Pp. 2, L 78 and 79).
Point 6: Based on 1), are the conclusions you presented always valid or it needs further studies over time (i.e. years subsequent to the sowing year). A comment on it should be appropriate and useful, for example at the end of discussion section or considered in conclusions.
In other words, is the (final) conclusion at line 334-335 valid/true or not for the years subsequent to the sowing year?
L 334 Consequently, AGDD is a feasible alternative to be used as a defoliation criterion for L. perenne and B. valdivianus pastures.
Response 6: In the final conclusion, we clarified that it is feasible under our experimental condition and, we invite other researchers to do the same in other environments and with other pasture species (Pp. 11, L 341, 342 and 343).
Thank you to the reviewer for the valuable contribution.
Reviewer 3 Report
Authors did a good job with the research design and presentation, and the paper brings a contribution to the management of cool-season and temperate forage.
Some suggestions below:
-line 31: ...the pastures that were submitted to...
-suggest:... the pastures allocated to...
Use consistent terms for DF:
line 33: , in this line you use highest, in line 35 you use shortest and largest. (be consistent with terms: highest vs largest).
-Consolidate information. And round the means based on standard error.
line 35: Suggest: (90 AGDD; 221 g kg-1 CP) to the largest DF (450 AGDD; 138 g kg-1 CP
-Line 50, suggest to add: In cool-season grasses, temperature is one of the main
-Line 69-71, Despite this... add...In Chile, growth dynamics... the pasture
-line 78: add at end "under Chilean conditions"
Line 86: add one sentence on agronomic implications.
Line 99: performed
line 156: Start discussion with the significant interaction...
line 159: delete sentence...do not report what was not significant
line 186, Table 1: check LGR? or LER
Author Response
Response to Reviewer 3 Comments
Point 1: -line 31: ...the pastures that were submitted to...
-suggest:... the pastures allocated to.... 

Response 1: Yes, we added the suggest (Pp. 1, L 32).
Point 2: Use consistent terms for DF.
Response 2: We check either manuscript and changed defoliation frequency for DF.
Point 3: line 33: , in this line you use highest, in line 35 you use shortest and largest. (be consistent with terms: highest vs largest).
Response 3: We appreciate a lot the suggestion, but we think that is better to use "Higher" for denoting quantities and "largest" or "shortest" to refer to defoliation frequencies.
Point 4: -Consolidate information. And round the means based on standard error. line 35: Suggest: (90 AGDD; 221 g kg-1 CP) to the largest DF (450 AGDD; 138 g kg-1 CP
Response 4: Changes were made in the approximation of the text units, but in the tables the values were left with the decimals (Pp. 1, L 36 and 37).
Point 5: -Line 50, suggest to add:In cool-season grasses, temperature is one of the main
Response 5: We added all the suggestions in L- 50 and now they are in Pp 2, L 51 and 52.
Point 6: -Line 69-71, Despite this... add...In Chile, growth dynamics...the pasture
Response 6: We added all the suggestions from L- 69 to L 71 and now they are in Pp 2, L 72 and 73.
Point 7: -line 78: add at end "under Chilean conditions"
Response 7: We added the suggestion of L- 78 and stayed in the same line.
Point 8: Line 86: add one sentence on agronomic implications.
Response 8: The suggestion of add the agronomic implications was considered (Pp 3, L 95).
Point 9: Line 99: performed
Response 9: The Word “performed” was deleted in the text (Pp 3. L 105).
Point 10: line 156: Start discussion with the significant interaction...
Response 10: Yes, we could start the discussion with the interactions that were significant, but we decided to start the discussion in the order that the main effects are shown in the results table.
Point 11: line 159: delete sentence...do not report what was not significant
Response 11: The sentence “do not report what was not significant” was deleted which was in Pp. 4, L 164.
Point 12: line 186, Table 1: check LGR? or LER.
Response 12: The correction of LER was made in table 1 (Pp. 6, L 193).
Thank you to the reviewer for the valuable contribution.